

# Quantitative analysis of rat adipose tissue cell recovery, and non-fat cell volume, in primary cell cultures

Floriana Rotondo[1,2], María del Mar Romero[1,2,3], Ana Cecilia Ho-Palma[1], Xavier Remesar[1,2,3], José Antonio Fernández-López[1,2,3] and Marià Alemany[1,2,3]

[1] Department of Biochemistry and Molecular Biomedicine, Faculty of Biology, University of Barcelona, Barcelona, Spain
[2] Institute of Biomedicine, University of Barcelona, Barcelona, Spain
[3] CIBER OBN, Barcelona, Spain

## ABSTRACT

**Background**. White adipose tissue (WAT) is a complex, diffuse, multifunctional organ which contains adipocytes, and a large proportion of fat, but also other cell types, active in defense, regeneration and signalling functions. Studies with adipocytes often require their isolation from WAT by breaking up the matrix of collagen fibres; however, it is unclear to what extent adipocyte number in primary cultures correlates with their number in intact WAT, since recovery and viability are often unknown.

**Experimental Design**. Epididymal WAT of four young adult rats was used to isolate adipocytes with collagenase. Careful recording of lipid content of tissue, and all fraction volumes and weights, allowed us to trace the amount of initial WAT fat remaining in the cell preparation. Functionality was estimated by incubation with glucose and measurement of glucose uptake and lactate, glycerol and NEFA excretion rates up to 48 h. Non-adipocyte cells were also recovered and their sizes (and those of adipocytes) were measured. The presence of non-nucleated cells (erythrocytes) was also estimated.

**Results**. Cell numbers and sizes were correlated from all fractions to intact WAT. Tracing the lipid content, the recovery of adipocytes in the final, metabolically active, preparation was in the range of 70–75%. Cells showed even higher metabolic activity in the second than in the first day of incubation. Adipocytes were 7%, erythrocytes 66% and other stromal (nucleated cells) 27% of total WAT cells. However, their overall volumes were 90%, 0.05%, and 0.2% of WAT. Non-fat volume of adipocytes was 1.3% of WAT.

**Conclusions**. The methodology presented here allows for a direct quantitative reference to the original tissue of studies using isolated cells. We have also found that the "live cell mass" of adipose tissue is very small: about 13 $\mu$L/g for adipocytes and 2 $\mu$L/g stromal, plus about 1 $\mu$L/g blood (the rats were killed by exsanguination). These data translate (with respect to the actual "live cytoplasm" size) into an extremely high metabolic activity, which make WAT an even more significant agent in the control of energy metabolism.

Corresponding author
Marià Alemany, malemany@ub.edu

## INTRODUCTION

White adipose tissue (WAT), which has been defined as the adipose organ (*Cinti, 2001*), is dispersed in a large number of locations, in which its basic energy storage activity is complemented by many other physiological functions (*Alemany & Fernández-López, 2006*). In any case, its main acknowledged role is to contribute to the defense of energy homoeostasis, helping to control glucose (*Sabater et al., 2014*), lipid (*Deschênes et al., 2003*; *Wang et al., 2016*), and amino acid (*Arriarán et al., 2015a*) metabolism overall. It is responsible for an important share of the control of whole body energy availability (*Hall, Roberts & Vora, 2009*; *Choe et al., 2016*), and acts as a platform for the immune system, being actively implicated in processes of protection and repair (*Parker & Katz, 2006*; *Dixit, 2008*). The complex (and varying) mixture of cell types in WAT depots largely determines and modulates these functions as part of its adaptability (*Vielma et al., 2013*; *Oishi & Manabe, 2016*).

Most of WAT volume is taken up by a relatively small number of large cells, the mature adipocytes, which are generally considered the genuine cells of this tissue and thus the main target for the fight against obesity (*Nawrocki & Scherer, 2005*). However, most of the adipocyte volume is filled by (triacylglycerol) energy reserves (*Kotronen et al., 2010*). This can be extended, obviously in similar proportions, (often higher than 80%) to the WAT/adipose organ taken as a whole. This is a variable but significant share of total body weight (5–50%) in humans and most animal phyla. The rest of WAT cells are loosely called stromal, despite most of them not being actually connective tissue cells (*Da Silva Meirelles et al., 2015*). In this text, we will use the general term "stromal cell" to refer to all WAT cells different from fat-laden adipocytes.

The stromal fraction of WAT is made up of immune system, stem, blood, endothelial, true stromal and other types of cells, with relevant functions in the maintenance of adipocyte energy homoeostasis (*Sadie van Gijsen et al., 2012*), defense (*Hill, Bolus & Hasty, 2014*), regeneration (*Domergue et al., 2016*), differentiation (*Gimble et al., 2011*; *Mitterberger et al., 2014*) and others (*Sumi et al., 2007*; *Takahara et al., 2014*). Many of these functions become critical under conditions of inflammation (*Lee, 2013*), changing the cell composition and overall WAT metabolism (*Lolmède et al., 2011*; *Cignarelli et al., 2012*). Adipocytes, despite their small numbers (but huge volume due to their fat stores), have been intensely studied as "representative" of WAT (*Leonhardt, Hanefeld & Haller, 1978*). To study their metabolic or regulatory capabilities, the cells are isolated from WAT masses and studied using primary (*Garvey et al., 1987*) or immortalized (*Tordjman, Leingang & Mueckler, 1990*) cell cultures. The information obtained is often taken as directly representative of WAT *in vivo,* in spite of the large number of factors that are known to rebut this excessively simplistic approach (*O'Brien et al., 1996*), including the ordeal of cell isolation (*Thompson et al., 2012*).

When dealing with WAT, the data obtained from most experiments is deeply conditioned by the methodology used, i.e., isolated cells, tissue pieces or slices, or *in vivo* functional analyses. Seldom can we obtain quantitative data which could be referred to the live tissue. Comparison of different locations, individuals, metabolic or pathologic conditions is severely hampered by the size of fat depots (*Cinti, 2001*; *Wronska & Kmiec, 2012*), the varying proportion of adipocyte/stromal cells (in fact, only when the latter are actually taken

into account (*Pasarica et al., 2009*) and the blood flow/oxygen and substrates' availability (*Mjös & Akre, 1971*). Quantification of adipocyte recovery from whole tissue samples, and the analysis of the proportion of "live" cell space in the tissue are necessary steps for direct comparison of data from different sources. Unfortunately, cell number is dependent on the method of quantification used, and is logically affected by cell volume. The proportion of fat in the tissue and cells also proportionally "reduces" the live-cell mass. This is further confounded by the direct estimation of cell numbers via DNA analysis which (at least in mammals) would not detect the number of erythrocytes, but would detect numbers of small hematopoietic cell (*Luche et al., 2015*) macrophages and lymphocytes (*Sell & Eckel, 2010*). The latter non-adipocyte populations would then be counted as "adipocytes," despite having a volume about $10^4$-fold smaller.

Referring cell or tissue experimental data to protein content may be a fair index for comparison, but the large presence (also deeply varying depending on location (*Alkhouli et al., 2013*)) of extracellular fibrous proteins, such as collagen (*Liu et al., 2016*) also modifies the quantitative evaluation of the metabolically active fraction of the tissue; this fraction is also deeply affected by obesity and inflammation (*Li et al., 2010*).

In the present study, we have devised a method for the estimation of actual recovery of viable adipocytes with respect to WAT mass based on the unique presence of large amounts of fat in them. We have also intended to present an estimation of the size of the metabolically active WAT cell mass with respect to the mass/volume of the tissue. We used, as reference, the epididymal WAT fat pads of non-obese healthy adult rats (to limit the known effects of inflammation on WAT cell profile). This location is considered to be one of the less metabolically active (*Arriarán et al., 2015b*), and is widely used for "representative" WAT adipocyte function for its size, easy dissection and absence of contamination by neighboring tissues.

## MATERIALS AND METHODS

### Rats and housing conditions

All animal handling procedures and the experimental setup were in accordance with the animal handling guidelines of the corresponding European, Spanish and Catalan Authorities. The Committee on Animal Experimentation of the University of Barcelona specifically authorized the procedures used in the present study.

Male Wistar rats (Harlan Laboratory Models, Sant Feliu de Codines, Spain), 18-week old, weighing $435 \pm 84$ g (mean, SD), were used after a 2-week acclimation period in a controlled environment. The animals were kept in two-rat cages with wood shards as bedding material, at 21–22 °C, and 50–60% relative humidity; lights were on from 08:00 to 20:00. They had unrestricted access to water and standard maintenance rat chow (Harlan #2014).

### Isolation of adipocytes

The rats were killed, under isoflurane anesthesia, at the beginning of a light cycle, by exsanguination from the exposed aorta, using dry-heparinized syringes. The rats were rapidly dissected, taking samples of epididymal WAT, used immediately for adipocyte isolation. This procedure followed, essentially that described by *Rodbell (1964)*. In short, tissue

samples were weighed, immersed in the digestion medium described below, and cut in small pieces with scissors. Samples were incubated, at 37 °C in a shaking bath for 60 min, with 2.5 volumes of Krebs-Henseleit buffer pH 7.4, containing 5 mM glucose, 0.1 µM adenosine (Sigma-Aldrich, St Louis, MO, USA) (*Honnor, Dhillon & Londos, 1985*), and 10 g/L lipid-free bovine serum albumin (Merck-Millipore, Billerica, MA USA). This was complemented with 3.5 mkat/L collagenase (LS004196, type I; Worthington Biomedical, Lakewood, NJ, USA). The collagenase-containing digestion buffer was prepared in the cold room (4 °C), and was used within 1 h.

At the end of the digestion process (carried at 37 °C), the suspensions were gently sieved using a double layer of nylon mesh hose (plain commercial sheer tight stocking; 90% polyamide, 10% elastomer, parallel woven with 15 den cylindrical single-filament threads; with approximate mean—flexible—pores in the range of 300 µm), which retained vessel fragments and (eventually) undigested tissue pieces. The smooth crude suspension of isolated cells was left standing for 5 min in stoppered polypropylene syringes (#SS+10ES1, Terumo, Tokyo, Japan), held vertically, at room temperature (22–24 °C). The adipocytes floated to form a defined upper layer. Then, the lower aqueous fraction was slowly drained off, capping again the syringe to retain the adipocytes. The cells were washed this way three times, using 2.5 volumes of the buffer each time. Before re-suspending the cells in it, the buffer was subjected to 30 s vortexing, to allow for equilibration with air oxygen. The final supernatant fraction contained intact adipocytes and a thin layer of free fat from broken cells. After the final washing, 400 µL aliquots of the cells' fraction were taken for incubation. The samples were slowly extracted from the central part of the adipocytes' layer, trying not to disturb the thin-floating lipid layer. The cells were manipulated and maintained at room temperature for a time as short as possible, and used immediately after the final washing.

Stromal cell space in the isolated cell suspension, used to relate their numbers and volumes to initial tissue weight, was considered the sum of the volume of the lower phase of adipocyte separation in the syringes, plus the volume of the adipocyte phase, to which the volume of adipocytes (calculated from cell numbers and volumes) was subtracted. Obviously, the first separation of adipocytes and stromal cells left a high number of the latter mixed with adipocytes. The three successive washings resulted in the presence (calculated) of, at most, 0.1% of the initial stromal cells in the final washed adipocyte fraction (down from an initial 7.3%). This assumption does not take into account stromal cells bound, retained or attached to the larger adipocytes.

### Estimation of the efficiency of adipocyte extraction

Practically all fat in WAT is limited to adipocytes. All types of cells contain lipids, mainly as membrane components; the small size of the combined mass and their density do not alter the cells' density and, consequently their buoyancy. A few types of cells, i.e., macrophages, foam cells and differentiating preadipocytes may contain sizeable amounts of fat, but they only appear under precise physiological conditions (foam cells, differentiating preadipocytes) and their numbers and size (and thus their combined content of fat) make their contribution small (negligible in the present case). All other cells do not have sufficient

lipid to generate enough floatability to allow their separation from the rest of cells by just standing—i.e., at $1 \times g$—for five minutes. We used this differential fat content to establish an approximate estimation of the efficiency of the digestion-extraction procedure for adipocyte isolation described above, simply by estimating the recovery of fat from the intact tissue to a preparation containing only viable functional cells.

A sample of just dissected WAT was divided in two parts, one was processed to obtain washed adipocytes as described above, and the other was divided in several aliquots, used to measure the water (dry weight after 24 h at 90 °C) and lipid content. To measure lipids, fragments of about 300 mg of intact tissue were weighed and extracted with trichloromethane: methanol (2:1 v/v) (*Folch, Lees & Sloane-Stanley, 1957*). The resulting values were used to establish the proportion of lipids in the intact tissue. Using this method as originally described, most of membrane lipids were not extracted (*Rose & Oklander, 1965*; *Eder, Reichlmayr-Lais & Kirchgener, 1993*), but the recovery of WAT-vacuole lipids (i.e., fat, essentially triacylglycerols) was quantitative. The weights of the lipids present in the fat layer on top of the cells' suspension (washed and essentially free of stromal cells, as explained above), and those of stromal cells' fraction and extraction debris were measured. The weight of the recovered adipocyte fraction and their water and lipid content were also estimated, thus obtaining the total weight of lipid present in the isolated adipocytes.

The density of WAT was estimated using tightly capped tubes, which were weighed both dry and completely full of deionized water at 20 °C. The net weight of water was used to calculate the volume of the tube. The process was repeated including weighed 300–500 mg pieces of intact WAT in the tubes and completely filling them with water (nevertheless, no different values were obtained using pieces of 200–1,000 mg). The difference in weight of the tubes with and without WAT samples allowed us to calculate the volume of the samples; their density was estimated from the volume and weight. Other samples of WAT were used to extract its lipid as described above. The density of the extracted lipid was estimated using the same procedure using cold-solidified fat samples.

The weight of lipid extracted from the adipocyte preparation was compared with the initial weight and the actual proportion of lipid present in the intact tissue, after discounting the weight of debris eliminated during the process of extraction. Lipid in the stromal cell fraction was negligible, statistically not different from zero.

## Measurement of isolated cell parameters

A known volume of the suspension of adipocytes was introduced in a Neubauer chamber (#717810 Neubauer improved bright line; Brand Gmbh, Wertheim, Germany). Using an inverted microscope, four fields (following a pre-established selection pattern) were photographed at low power (Fig. 1). Four samples of each adipocyte suspension were inspected, taking 16 photographs from each. Cells were identified, counted, and their diameters analyzed (under the conditions used, all cells adopted a spheroid form), using the *FIJI ImageJ* software (http://imagej.nih.gov/ij/), following a simple procedure (*Baviskar, 2011*). The data were computed (range, mean and SD for diameter, cell volume and number, including their combined volume). In this experiment, the final range of counted cells (mean, SD) was $96 \pm 10 \,\mu\mathrm{m}$ in diameter (when assuming the form of a sphere), i.e., 475

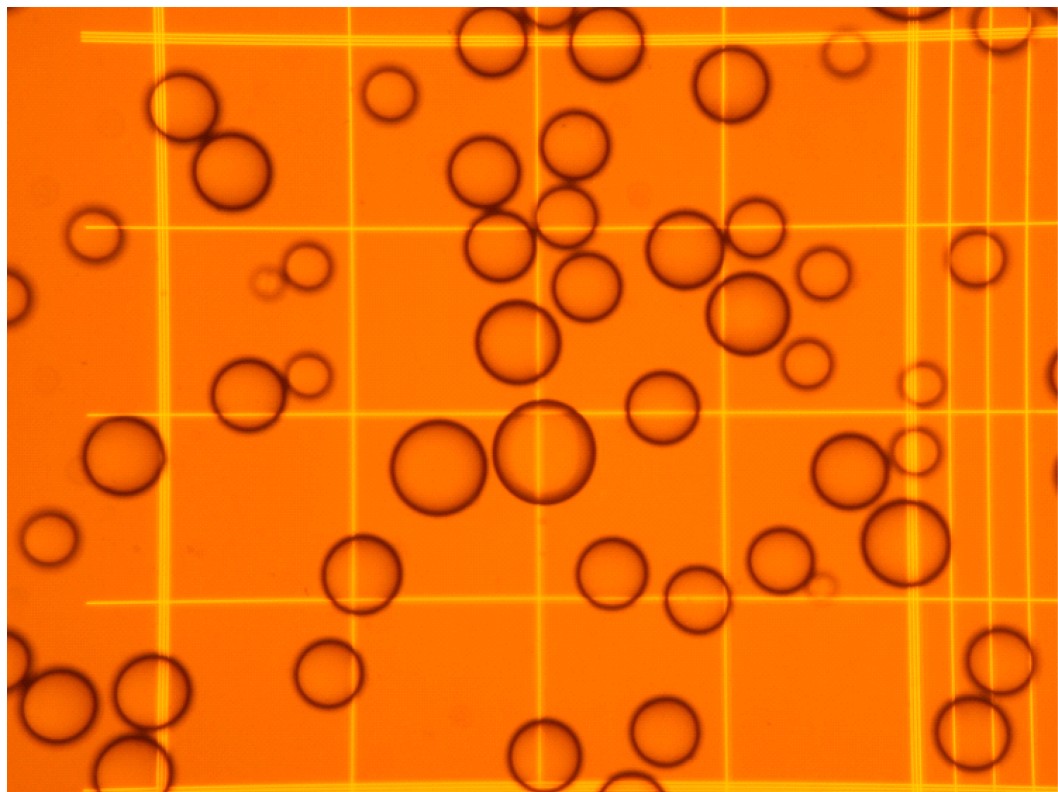

**Figure 1** **Representative microphotography of an adipocyte preparation observed at the microscope using a Neubauer chamber.** The squares in the grid have a width of 250 μm, and correspond to a volume of 6.25 nL.

± 147 pL in volume. Figure 2 shows a representative example of the range of cell sizes obtained using this procedure on epididymal WAT.

Non-nucleated cells (essentially red blood cells: RBCs) were identified by their smaller size (in the fL range) using the Scepter 2.0 cell counter (EDM Millipore Corp, Billerica, MA USA) hand-held cell sizer. Total stromal cells, (i.e., including RBCs) were analyzed for each sample using two different cell-range tips for the Scepter: Sensor 40, for 3–18 μm particles' size (PHCC40050; Merck Millipore, Darmstadt, Germany) and Sensor 60, for 6–36 μm particles' size (PHCC60050; Merck Millipore). The data for both ends of the superimposed size graphs were taken as final values, and those in the overlapping zone were used taking in both series of data against diameter. After the data were arranged, the measured volumes were plotted and the data were statistically analyzed.

Using stromal cell fraction samples from all rats tested, a cytometric flow analysis (Fig. 3) was performed to distinguish the proportion of small non-nucleated cells (i.e., red blood cells) from those nucleated and either dead or viable. The analyses were done using a FacsAria I SORP sorter (Beckton-Dickinson, San Jose, CA, USA). The cells were stained with propidium iodide (Sigma-Aldrich) and Syto-13 (Life Technology, Thermo-Fisher Scientific, Waltham, MA USA) used to estimate the proportion of non-nucleated red blood cells in the samples as a percentage of total stromal cells. We used this value to estimate

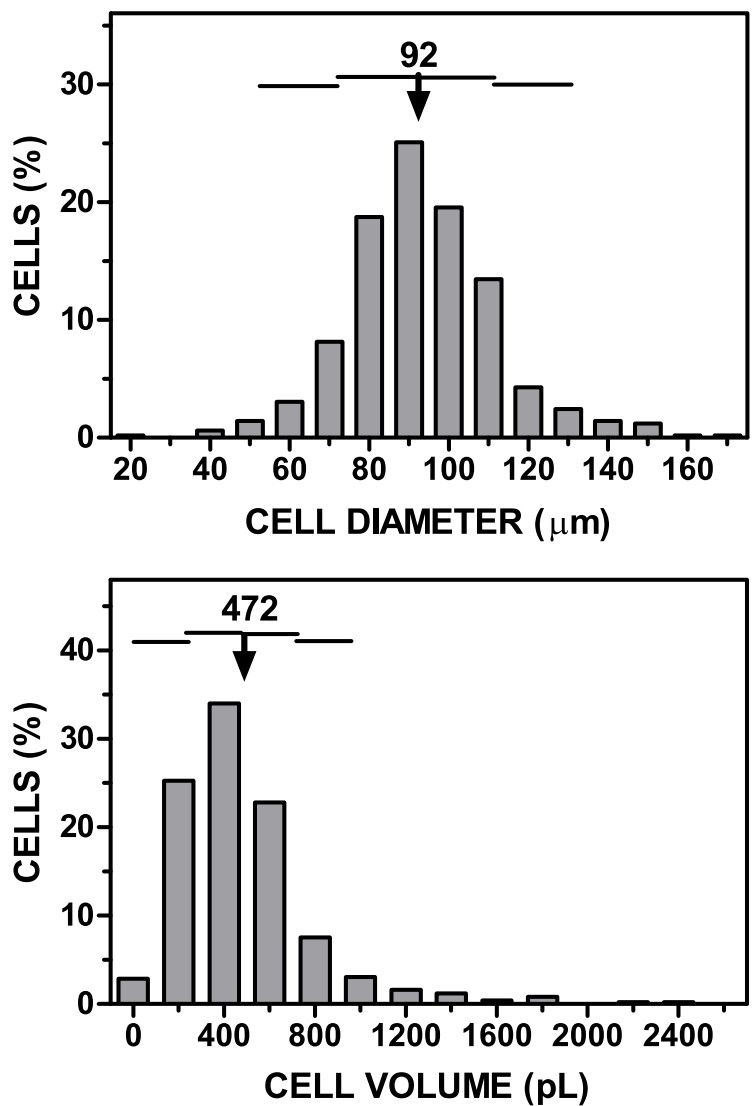

**Figure 2** Representative graph of cell size (diameter, volume) *vs*. cell numbers representation obtained applying the cell extraction procedure described in the text to a sample of epididymal adipose tissue. The data have been grouped to facilitate the presentation. The arrow (and the number above) represent the mean cell diameter and volume. The horizontal lines represent each one the extent of one SD.

the presence of blood cells in the whole tissue and stromal cell counts, incorporating these data in the calculations.

## Cell viability

We analyzed the functionality of the cells checking their metabolic integrity along a 2-day incubation study. We used 12-well plates (#CLS3513 Costar; Sigma-Aldrich) filled with 1.7 ml of DMEM (#11966-DMEM-no glucose; Gibco, Thermo-Fisher Scientific, Waltham, MA, USA), supplemented with 30 mL/L fetal bovine serum (FBS, Gibco). The medium (*Romero et al., 2015*) also contained 25 mM hepes (Sigma-Aldrich), 2 mM glutamine (Lonza Biowhittaker, Radnor, PA, USA), 1 mM pyruvate (Gibco), 30 mg/mL delipidated

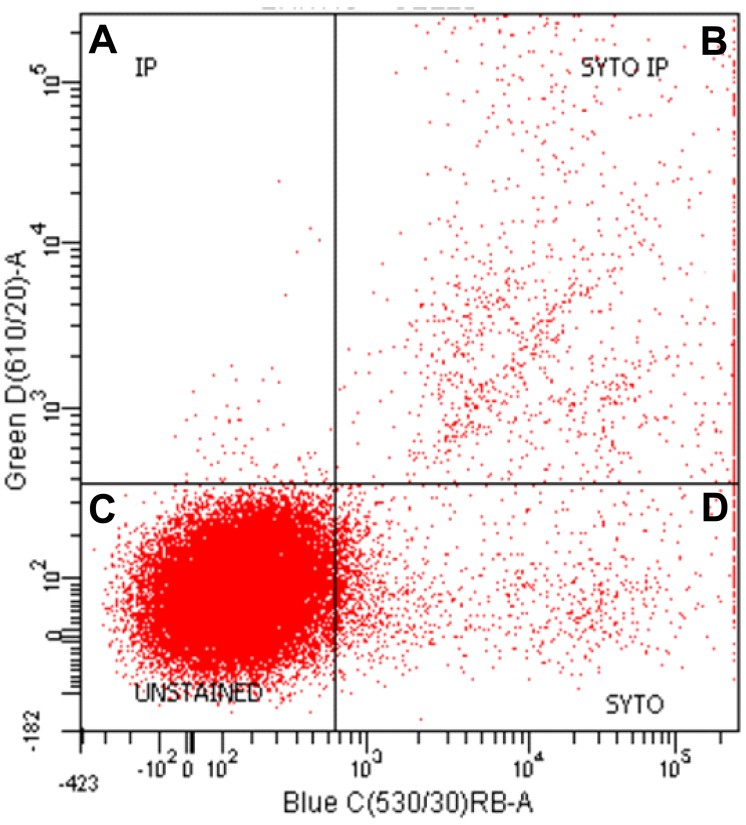

**Figure 3** **Representative graph of flow cell analysis of stromal fraction of epididymal rat WAT to discriminate nucleated from non-nucleated cells.** Both propidium Iodide (IP) and Syto-13 (SYTO) bind DNA-positive and double positive particles (i.e., cells). The dots in the (A), (B) and (D) correspond to nucleated stromal cells; dots in (C) show the unstained cells, largely corresponding to the high proportion of erythrocytes.

bovine serum albumin (Millipore Calbiochem, MA, USA), 100 U/mL penicillin and 100 mg/L streptomycin (Gibco). Adenosine (Sigma-Aldrich) 100 nM was also added to help maintain the integrity of the cells. D-glucose (7 mM) was added as substrate. Each well received 400 μL of the adipocyte suspension (a second 100 μL aliquot was taken simultaneously to determine the adipocyte content in the well), thus completing a final volume of 2.1 mL. Under these conditions, the cells floated freely (as spheres) and tended to accumulate on the surface of the well. The cells were incubated at 37 °C in an incubation chamber ventilated with air supplemented with 5% $CO_2$, which gave a theoretical $pO_2$ of 20 kPa (i.e., 0.2 mM of dissolved $O_2$) (*Romero et al., 2015*). The calculated $pCO_2$ was in the range of 5 kPa, corresponding to 1.7 mM of dissolved $CO_2$. The cells were incubated for 24 h or 48 h without any further intervention. At the end of the experiment, a sample of the well contents was used to determine the number of cells. Then, the cells were harvested and the medium was extracted and frozen.

The incubation medium was used for the estimation of glucose, using a glucose oxidase-peroxidase kit (#11504; Biosystems, Barcelona, Spain) to which we added 740 nkat/mL mutarrotase (porcine kidney, 136A5000; Calzyme, St Louis, MO, USA) (*Oliva et al., 2015*).

Lactate was measured with kit 1001330 (Spinreact, Sant Esteve d'en Bas, Spain), glycerol was estimated with kit #F6428 (Sigma-Aldrich); NEFA were measured using kit NEFA-HR(2) (Wako Life Sciences, Mountain View, CA, USA).

## Calculations

A critical factor in the development of this procedure was to keep track of all weights/volumes and incorporate into the calculations all aliquots extracted for testing (i.e., glucose or lactate levels). All data were introduced in a spreadsheet in which the volumes were justified with a (pipetting) error of $\pm 3\%$. When possible, or when no other avenue was available, volumes were estimated from differential weights and the application of the densities calculated as described above.

The calculations used to determine the cell parameters, adipocyte recovery and WAT cell distribution are described in the Tables, presenting the original experimental data along with the derived or calculated data, as well as the formulas used for their estimation.

Statistical analyses were carried out using the Prism 5 Program (Graphpad Software Inc., La Jolla, CA, USA). Statistical differences between groups of data were determined with the unpaired Student's *t* test.

## RESULTS

### Analysis of the recovery of adipocytes from intact epididymal WAT

Table 1 shows the main experimental data for the quantitative analysis of free isolated adipocyte yield from just-dissected epididymal WAT. Both weight, water and fat content, as expected, showed little variation. The suspension obtained after collagenase digestion was estimated by weight, as were the floating fat layer and the debris retained in the nylon mesh. The number, and mean volume of intact adipocytes was also fairly uniform. The number of free (i.e., unattached to adipocytes) stromal cells was 17-fold higher than that of adipocytes, but almost 3/4ths of them were just red blood cells. All stromal cells had cell volumes in the range of $10^{-4}$ of those of adipocytes. The volumes of all stromal cells, including erythrocytes were measured after separation via high-speed centrifugation, which may have altered their original shape and volume, a treatment that the large adipocytes could not endure.

All tables contain a first column, labeled #, in which a letter and number are given to each row (or datum). These references are later used, in Tables 2–6 to present the origin of the data and the calculations done using the experimental data.

Table 2 presents the calculations (largely based on the data in Table 1) used to determine the recovery of viable isolated adipocytes from the intact tissue sample. Since all experimental data referred to weight (its measurement was several-fold more precise than volumetric measurements, especially those implying solids—such as cells—in suspension and mixed-phase systems) the main column of data is that indicated by weights, and have been referred to mg in 1 g of initial tissue. These values were converted to volumes using the densities experimentally measured for fat and tissue shown in Table 1. The third column shows the origin of the data and the calculations used to obtain the corresponding values.

**Table 1  Results obtained from the collagenase digestion of rat epididymal WAT and the analysis of the tissue and fractions of tissue obtained in the process of separation of viable isolated adipocytes.** The data presented as mean ± SD are direct experimental results obtained from four different rats.

| # | Parameter | Units | Values |
|---|---|---|---|
| A1 | Epididymal WAT weight | g | 4.32 ± 0.44 |
| A2 | WAT fat content | mg/g | 869 ± 15 |
| A3 | WAT water content | mg/g | 45 ± 6 |
| A4 | Adipocyte suspension (digested tissue) | g | 4.78 ± 0.86 |
| A5 | Floating fat derived from broken adipocytes | mg | 105 ± 96 |
| A6 | Intact adipocytes suspension (A4–A5) | g | 4.67 ± 0.85 |
| A7 | Fat in the intact adipocytes suspension | mg/g | 537 ± 199 |
| A8 | Total fat in the intact adipocytes suspension | g | 2.51 ± 1.06 |
| A9 | Water in the intact adipocytes suspension | mg/g | 287 ± 68 |
| A10 | Recovery of intact adipocytes | cells × $10^6$ | 5.82 ± 3.06 |
| A11 | Adipocyte mean volume | pL | 475 ± 147 |
| A12 | Extraction debris mass (dry weight) | mg | 356 ± 13 |
| A13 | Number of total stromal cells freed | cells × $10^6$ | 103 ± 45 |
| A14 | Stromal cells' mean volume | fL | 96.6 ± 43.0 |
| A15 | Red blood cells (proportion of A13, total stromal cells) | % | 71.4 ± 8.5 |
| A16 | Red blood cells' mean volume | fL | 25.9 ± 1.1 |
| dt | Intact WAT density | g/mL | 0.940 ± 0.013 |
| dl | WAT fat density | g/mL | 0.922 ± 0.022 |

**Table 2  Analysis of the effectivity of the adipocyte isolation procedure used based on the analysis of lipid distribution, from intact tissue to the final preparation of adipocytes.** The data are mean values calculated from the experimental data in Table 1. The column "calculations" explains the data used in each case. Volumes were calculated with $dt$ or $dl$ (Table 1) when applied to tissue ($V = W/dt$) or lipid ($V = W/dl$), where $W$ is weight (in g) and $V$ volume (in mL). In the calculations marked ($W$ and $V$), the values were calculated directly from weights and volumes, i.e., not applying the density factors.

| # | Parameter | Weight mg/g intact WAT | Volume µL/g intact WAT | Calculations |
|---|---|---|---|---|
| B1 | Intact epididymal WAT | 1,000 | 1,064 | |
| B2 | Extraction debris (dry weight) | 83 | 88 | (A12 × B1)/A1 |
| B3 | WAT fat content | 869 ± 15 | 943 | A2 |
| B4 | WAT mass minus debris | 917 | 976 | B1 − B2 ($W$ and $V$) |
| B5 | WAT fat content corrected by debris | 797 | 865 | (B3 × B4)/B1 ($W$ and $V$) |
| B6 | Lipid, from broken adipocytes, in the fat layer | 24 | 26 | (A5 × B1)/A1 |
| B7 | Total WAT fat in the extracted adipocytes | 773 | 838 | B5 − B6 ($W$ and $V$) |
| B8 | Total fat in the intact adipocytes recovered | 581 | 630 | (A8 × B1)/A1 |
| B9 | Total fat in the adipocytes recovered (intact or broken) | 605 | 657 | B6 + B8 ($W$ and $V$) |
| B10 | Fat loss during adipocyte isolation | 192 | 208 | B5 − B9 ($W$ and $V$) |
| B11 | Percentage of adipocyte fat recovery | 75.9 | – | (B9/B5) × 100 |
| B12 | Percentage of adipocytes (fat) lost in the fat layer | 3.1 | – | (B6/B5) × 100 |
| B13 | Percentage of intact adipocytes (expressed as fat) in the final preparation | 72.8 | – | (B8/B5) × 100 |

The detailed calculations of the efficiency of adipocyte recovery can be seen on Tables 1 and 2. We assumed that practically all WAT fat was present in the adipocyte fraction, essentially in adipocytes, since membrane lipids were not extracted with the procedure used (*Rose & Oklander, 1965*; *Remesar et al., 2015*), the eventual presence of fat in stromal cells went undetected and, in any case, could not represent a significant amount of material given the combined volume of these cells and their density. Consequently, all the fat present in the final intact adipocyte preparation should correspond to that of adipocytes, since free fat was measured and removed, and there were no other fat-carrying cells in the system in mass and/or numbers sufficient to alter the results, and neither membrane lipids could interfere in a significant way. Our previous work provides additional calculations that further support this conclusion (*Remesar et al., 2015*). Thus, we could equate the losses of fat (with respect to intact tissue) with losses of adipocytes. These losses were found to be significant, and the manipulation of the cells resulted in additional cells breakup. Under the conditions described, the collagenase incubation and extrusion through the nylon mesh resulted in a loss of about 24% of the cells (in fact, losses of fat), and the washings of the isolated cells added an additional loss in the range of 3%, which resulted in a recovery of about 73% of intact functional cells in the final adipocyte preparation, used for incubations, and referred to intact WAT (Table 2).

## Isolated adipocyte viability

The viability of the cells obtained was high in the final preparation, with a negligible number of cells broken. The incubation of cells (about 700,000 per well) resulted in a loss of cells of approximately 4% in the first 24 h and an additional 9% in the second 24 h period. Consequently, the cells were viable and remained functional for 2 additional days in primary culture. The rate of glucose uptake (and metabolic utilization) per cell increased significantly in the second day of incubation (Fig. 4). However, the lactate efflux rates were maintained. Glycerol efflux rate also rose several fold in the 24–48 h period, maintaining, in the end, a much higher efflux rate than that of NEFA, which attests to its mainly glycolytic origin (parallel to the increase in glucose uptake and the maintenance of lactate production). However, the sole presence of NEFA proves that lipolysis was clearly present in the second day, probably as a consequence of the loss of about half of the glucose initially present in the medium (i.e., decreasing its availability to support cell metabolism). The higher rates of glycerol efflux in comparison with those of NEFA also support the finding that most of glycerol was not of lipolytic origin (*Smith, 1972*; *Romero et al., 2015*), since then the reverse would be true. In any case, the data prove that metabolic activity (at least glucose uptake, glycolysis to lactate, glycerogenesis and lipolysis) were fully functional in the 48 h period studied, in fact increasing during the second day of incubation.

## Analysis of WAT cell type distribution and proportions, cumulative volumes

Table 3 shows the calculations derived from the data of Table 1 to obtain an approximate estimation of the combined proportions of tissue volume filled by the three main types of cells we were able to discriminate: adipocytes, nucleated stromal cells and red blood cells.

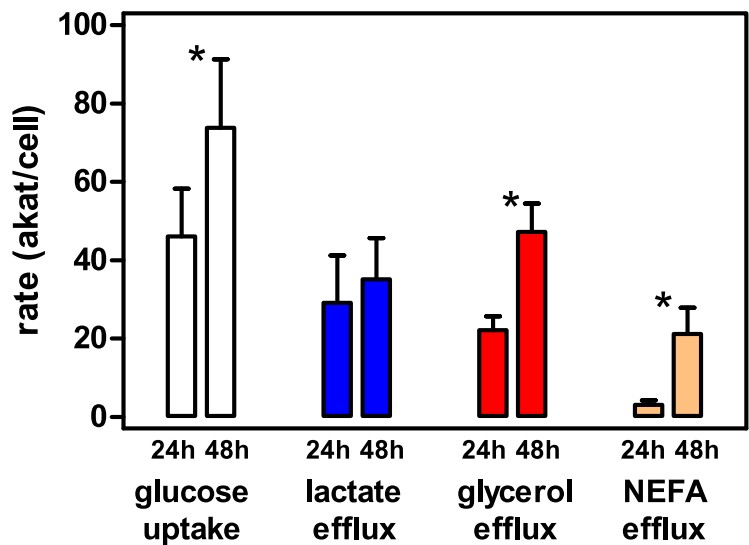

**Figure 4 Metabolic activity of epididymal WAT adipocytes in primary culture at 24 h and 48 h of incubation.** The data represent the mean ± SD of four different rats (triplicate wells). The data are presented as rates of uptake (glucose, white bars), or efflux to the medium (lactate, blue bars, glycerol red bars and NEFA beige bars), in concordant units (akat/cell) to facilitate comparisons. Statistical significance of the differences between 24 h and 48 h data: an asterisk * represents a $P < 0.05$ difference (Student's $t$ test).

**Table 3 Calculation of the volumes of cells from rat epididymal WAT.** Data calculated using the experimental results presented in Tables 1 and 2.

| # | Parameter | Units | Values | Calculations |
|---|-----------|-------|--------|--------------|
| C1 | Adipocytes in WAT | cells × $10^6$/g WAT | 1.85 | (B5/B1) × (A10/A8) |
| C2 | Combined volume of WAT adipocytes | μL/g WAT | 878 | (A11 × C1)/$10^6$ |
| C3 | Stromal cells in WAT | cells × $10^6$/g WAT | 23.9 | A13/A1 |
| C4 | Red blood cells in WAT | cells × $10^6$/g WAT | 17.0 | (C3 × A15)/100 |
| C5 | Nucleated stromal cells in WAT | cells × $10^6$/g WAT | 6.8 | C3 − C4 |
| C6 | Total volume of stromal cells in WAT | μL/g WAT | 2.3 | (C3 × A14)/$10^9$ |
| C7 | Total volume of red blood cells in WAT | μL/g WAT | 0.44 | (C4 × A16)/$10^9$ |
| C8 | Total volume of nucleated stromal cells in WAT | μL/g WAT | 1.87 | C6 − C7 |
| C9 | Mean nucleated stromal cell volume | fL | 273 | (C8/C5) × $10^9$ |

The total mass of adipocytes was scaled up to the tissue volume from the measured data of mean adipocyte volume and its numbers (estimated from tissue and isolated cells' fat content). Adipocytes constituted almost 0.88 mL/g WAT volume. Total stromal cells and erythrocytes' volumes were, likewise, calculated from their mean cell size and numbers, scaled up to the volume of 1 g of intact WAT. Despite their larger numbers, the combined total volume of all stromal cells accounted for a little more than 2 μL/g WAT.

Using the adipocyte fat content and its volume (both referred to 1 g of tissue weight minus debris), as shown in Table 4, we obtained an approximate estimation of the "live cell mas" of adipocytes in epididymal WAT. This volume included all the cell organelles, systems and cytoplasm, since the fat vacuole volume corresponds to the fat content, estimated from tissue mass and its direct measurement of fat content. The total cell volume, only slightly

**Table 4   Calculation of the non-fat cell volume of adipocytes in rat epididymal WAT.** Data calculated using the experimental data presented in Tables 1–3.

| # | Parameter | Volume (μL/g WAT) | % of total cells volume | Calculations |
|---|-----------|-------------------|-------------------------|--------------|
| D1 | Total volume of adipocytes in 1 g of WAT | 878 | 100 | C2 |
| D2 | Total fat volume in 1 g of WAT | 865 | 98.5 | B5 |
| D3 | Non-fat adipocyte cell volume in 1 g of WAT | 13 | 1.5 | D1 − D2 |

**Table 5   Distribution of cell types by volume and number in rat epididymal fat.** Data calculated using the results presented in Tables 1–4.

| # | Parameter | Volume μL/g WAT | % of WAT volume | Cells ($10^6$/g WAT) | % of WAT cells | Calculations |
|---|-----------|-----------------|-----------------|----------------------|----------------|--------------|
| E1 | Initial WAT weight (minus debris) | 976 | 100.0 | | | B4 |
| E2 | Adipocytes | 878 | 90.0 | 1.85 | 7.2 | D1 |
| E3 | Red blood cells | 0.44 | 0.05 | 17.0 | 66.2 | C7 |
| E4 | Nucleated stromal cells | 1.87 | 0.19 | 6.84 | 26.6 | C8 |
| E5 | Total cells | 880 | 90.2 | 25.7 | 100.0 | E2 + E3 + E4 |
| E6 | Extracellular space | 96 | 9.8 | | | E1 − E5 |
| E7 | Fat | 865 | 88.6 | | | D2 |
| E8 | Total stromal cell volume | 2.3 | 0.24 | | | C6 |
| E9 | Total nucleated cell volume | 880 | 90.2 | | | E2 + E4 |
| E10 | Adipocyte non-fat cell volume | 13.0 | 1.3 | | | D3 |

larger, was calculated from another set of data: cell counting and mean volumes, tracing the cell losses from those of fat. The small difference between both entities was in the range of 1.5% of the cell volume, and taken as such, despite the wide margin of error and the small number of animals used to calculate this mean value, it represents a very small proportion of the whole tissue, which magnifies its active metabolic performance.

Table 5 shows the global distribution of epididymal WAT volume and the space taken up by the three types of cells analyzed. Adipocytes took up 90% of the tissue volume (excluding the "debris," largely vessels and other structures or undigested tissue), but their number was only 7% of the total number of cells. Nucleated stromal cells hardly took 0.2% of the volume but accounted for 27% of the cells. Red blood cells were the most abundant, 66% of numbers, but their space was only 0.05%, a value that roughly corresponds to 1 μL of whole blood per g of WAT (the rats were exsanguinated, thus this is a residual tissue blood volume). Cells did not occupy all the tissue space, since about 10% of the tissue volume was extracellular space (interstitial and vascular). Fat alone filled 89% of the tissue space.

Table 6 summarizes the mean characteristics of the adipocytes extracted from rat epididymal WAT. Their estimated non-fat cell volume was in the range of 13 pL, much larger than the 273 fL of nucleated stromal cells and the 26 fL of red blood cells (Table 1). Adipocytes' "live cell volume" was 48× higher than nucleated stromal or 500× higher than red blood cells. But their complete volume (i.e., including the single fat vacuole) was, respectively, 1,700× and 18,000× larger. The combined non-fat adipocyte volume was (Table 5) about one order of magnitude higher than that of nucleated stromal cells. Thus,

**Table 6  Characteristics of the adipocytes isolated from rat epididymal adipose tissue.** Data calculated using the results shown in Tables 1–3.

| # | Parameter | Units | Values | Calculations |
|---|-----------|-------|--------|--------------|
| F1 | Lipid content | mg/g | 797 | B5 |
| F2 | Cell lipid weight | ng/cell | 431 | $B5/(C1 \times 10^6)$ |
| F3 | Cell lipid volume | pL/cell | 468 | $(B5/C1) \times 10^6/dl$ |
| F4 | Cell mean volume | pL/cell | 475 | A11 |
| F5 | Non-fat cell volume | pL/cell | 13 | F4 − F3 |

despite their lower numbers, the mass of "live-cell material" of adipocytes remains the main active component of WAT at least using these gross comparison tools.

## DISCUSSION

Probably, the most striking conclusion of the present study is the very small proportion of "live cell matter" found in epididymal WAT of normal young adult rats. Fat stores take up an inordinate amount of the tissue space, the interstitial space found is close to that described in previous reports and is in the range of other tissues (*Robert & Alemany, 1981*). However, after excluding the inert fat deposits, the remaining "cell material" accounts for about 1.5% of the total tissue mass, which seems very little even in relation to the assumedly limited metabolic activity of the tissue.

The data and viability of cells obtained with our customized version of the *Rodbell (1964)* method for isolation of adipocytes reflect a specific experimental condition, and their absolute values are obviously subjected to a number of possible modifying conditions, such as small changes in the conditions of extraction, the length of incubation, the inflammatory condition of WAT, the location of WAT depots, and the age, mass of WAT and sex of the animals used. Primary adipocytes may be incubated for long periods without loss of response to hormonal or paracrine stimuli (*Marshall, Garvey & Miriam, 1984*; *Fain & Madan, 2005*; *Giovambattista et al., 2006*). The lineal response to excess medium glucose producing lactate for up to 48 h is comparable to that described previously by us in 3T3L1 cells (*Sabater et al., 2014*). The increased secretion of glycerol and NEFA during the second day of incubation attest not to a loss of metabolic response and viability but to a change in the mechanisms of control of substrate efflux; these results agree with the known glycerogenesis and limited lipolysis of adipocytes when exposed to glucose (*Romero et al., 2015*).

It is well known that adipose tissue presents considerable difficulties to work with, the main problem being the dilution of cell proteins, RNA and DNA, as well as its wide variation in almost any parameter, largely attributed to the space occupied by huge fat stores. Evidently, this is not new, but the actual quantification, albeit approximate, of this entity is. The results may seem perhaps extreme, but the combined volume of fat (we often measure the weight, not the volume of fat depots) and extracellular space (i.e., plasma, and interstitial space) markedly limit the possible volume of the sum of blood cells, nucleated stromal cells and adipocyte non-fat cell volume. These considerations support, at least the range of "live cell" volume we have presented here for WAT. It is obvious

that the data calculated from the actual experimental results is only an approximation to the real values of this "live-cell" volume of adipose tissue cells. However, the data involved: percentage of fat in the intact tissue, and the combination of mean cell volume and number of adipocytes yield very close figures, with a small difference in cell size over vacuolar fat size. The different origin of the data, plus the use of different animals to get the means (the individual variability gave too much dispersion), decided us to work with experimentally-derived mean values to diminish the noise or clutter of individual data on the calculated/derived parameters. In previous works, we have proven the remarkable metabolic activity of the sum of WAT depots (i.e., taken as adipose organ) (*Arriarán et al., 2015b*; *Arriarán et al., 2015c*), especially its considerable glycolytic capability (under normoxic conditions) (*Arriarán et al., 2015c*; *Romero et al., 2015*), which adds to its known ability to store fatty acids taken from plasma lipoproteins (*Garfinkel, Baker & Schotz, 1967*; *Wang et al., 2016*) or synthesized from glucose (*Guerre-Millo, 2003*). Its important contribution to amino acid metabolism (*Arriarán et al., 2015a*), second only to liver (*Agnelli et al., 2016*; *Arriarán et al., 2016*), supports the long-proposed active WAT implication in energy and intermediate/substrate metabolism (*Cahill, 1962*). The data presented here only compound the puzzle, since the actual mass of cells doing the work is only a small fraction of the tissue, much lower than usually assumed. This small number of cells (including the stromal nucleated cells) is able to produce a large number of signaling cytokines (*Gerner et al., 2013*; *Wisse, 2004*), hormones (*Killinger et al., 1995*; *Stimson et al., 2009*) and maintain an active capacity to defend (immune system) (*Chmelar, Chung & Chavakis, 2013*), and repair or regenerate (i.e., stem cells) (*Ogura et al., 2014*) tissues. Compared to liver, which cell volume is upwards of 75% of its volume, the 50-fold lower proportion of WAT "live cell" volume has to show a much higher metabolic activity to be able to carry out the large number of functions and active metabolism that we keep discovering in recent times in WAT. The actual quantitation of the mass of adipocyte cytosol and its correlation with metabolic activity is a study worth carrying out, to definitively establish that WAT cells metabolism is extremely active, and not a dump for excess energy.

Surprisingly, the most abundant cells found in WAT were red blood cells, which accounted for roughly two thirds of the total. The volume of red blood cells was the approximate equivalent to about 1 µL of blood per g WAT, lower than previously published data using $^{65}$Fe-labelled red blood cells (*Robert & Alemany, 1981*). Probably, the blood figure will be higher *in vivo*, since the rats were killed by exsanguination, so that most of the blood was drained. Consequently, we can assume that *in vivo,* WAT blood content may justify a hefty proportion of the tissue cells.

For operative methodological simplicity, we have analyzed all non-adipocyte cells ("stromal") as a single entity, but we have considered apart, independently, red blood cells, first for their relatively large proportion, and second because of their limited metabolic activity (and absence of nuclei). Nevertheless, the combined volume of the nucleated stromal cells was, again, smaller than expected. We are reasonably certain that the methodology used accounted for all free tissue cells in this fraction, since only low-density cells (i.e., adipocytes, and—probably—differentiating preadipocytes) (*Grégoire et al., 1990*) were separated by the low centrifugation force used. Our stromal cell data are difficult to compare with

the large number of studies available that analyze WAT cell populations under different metabolic conditions, since in practically all cases, the studies are not quantitative, neither referred to initial tissue mass, and are usually centered on preadipocytes (*Grégoire et al., 1990*), macrophages (*Makkonen et al., 2007*), vascular (*Kajimoto et al., 2010*) or other specific cell types (*Villaret et al., 2010*). In addition, most data on WAT adipocyte counts were done in fixed and stained WAT histologic cuts, where, usually, only section areas (of adipocytes) are taken into account, irrespective of the level of the cell at which they have been sliced and then estimated.

The ever-present problem of lipid droplets in cell suspensions has been partially solved in this case by letting them coalesce in a lipid layer before counting adipocytes in micropho-tographs. Nuclear staining may open new possibilities for counting, but the probable presence of other cells attached to adipocytes (i.e., not removed by the washings) and the need to maintain the integrity of the cells for sizing has prevented the use of this approach in the present study.

We expected to find larger numbers of stromal cells, obviously more than blood cells, be-cause this relatively small part of the tissue is responsible for a large number of its metabolic functions and control responses as explained above, and is subjected to considerable variability related to its location and to inflammation (*Cildir, Akincilar & Tergaonkar, 2013*; *Villaret et al., 2010*). In any case, adipocytes remain by large (percentage of WAT volume either counting the fat vacuoles or not) the main component of WAT cell populations, but this primacy was lost when considering the numbers of cells.

One of the critical points this study tried to address was the efficiency of viable cell isolation from freshly dissected WAT and the maintenance of their functions for up to two days of incubation. The cell separation method we used is standard, and so widely used that seldom the source is cited, ensuring a fair recovery of the delicate adipocytes with minimal losses. We quantified these losses, and found that the recovery was initially close to 76% of the cells initially present in the tissue; but incubation resulted in the additional loss of significant (albeit relatively small) numbers of cells. In any case, we presented a method that allows the establishment of a quantitative relationship between the numbers of functional cells obtained with respect to the initial tissue mass, in the range of 73%. The data refer to viable cells, able to take up glucose from the medium, glycolyse it to produce lactate, and synthesize glycerol, part of which (as attested by the production of NEFA) was the product of lipolysis. In fact, the data presented show a marked increase in the efflux of glycerol and NEFA during the second day of incubation, at the expense of higher glucose uptake, proof that the cells were not losing functionality during the 24–48 h incubation, but increasing their utilization of glucose, which was comparable to that of 3T3L1 adipocytes (*Sabater et al., 2014*). The maintenance of function of adipocytes obtained with the *Rodbell (1964)* method, as is our case, has been repeatedly tested for periods of two days (or longer) in a wide variety of metabolic pathways and response to hormonal or chemical stimuli (*Marshall, Garvey & Miriam, 1984*; *Fain & Madan, 2005*; *Giovambattista et al., 2006*). However, the analysis of recovery was based essentially on the analysis of lipid in all fractions, so that the measurement of volumes (or weights) was critical and introduced a number of factors to be considered for success. First, all cells

floating in the buffer after treatment with collagenase and separation of debris (i.e., low density, and preferentially of large volume), were considered adipocytes. Just leaving the cells standing (i.e., centrifugation at $1\times g$) 5 min prevented pressure-caking of adipocytes and their breakage, but allowed a uniform distribution of smaller stromal cells between both phases. This was no problem for their estimation (numbers and volumes), but introduced a possible source of error when using isolated adipocytes for metabolic analysis, since the nucleated stromal cells remained a significant fraction of the crude adipocyte suspension. Three washings resulted in the loss of about 3% of adipocytes, but theoretically removed almost all non-attached stromal cells, down to a negligible proportion of the initial stromal cells content in the adipocyte fraction. The numbers and volumes of adipocytes found were in the range of those described in the literature (*DiGirolamo & Owens, 1976*; *Francendese & Digirolamo, 1981*). In addition, the cell volumes estimated, combined with the numbers of cells measured accounted for almost all the tissue space available, which is, in itself, an internal check that our calculations and estimations were essentially correct.

## CONCLUSIONS

The methodology presented here for the estimation of adipocyte recovery allows for a direct quantitative reference to the original intact tissue of studies with isolated cells. This way, the cultured cell data can be used as an approximation to metabolic activity and function related to whole organism.

We have presented proof that the "live cell mass" of adipose tissue is very small. This fact, translates into an extremely high (with respect to the actual "live cytoplasm" size) metabolic activity to justify the overall activity of WAT in glucose-fatty acid relationships, but also in amino acid metabolism. These data justify that comparison of epididymal WAT, often considered the less metabolically active part of the adipose organ, with more metabolically relevant tissues such as liver should take into account these quantitative data, which make WAT an even more significant agent in the control of energy metabolism.

## ACKNOWLEDGEMENTS

Thanks are given to Dr. Marta Camps for her help with practical aspects cell isolation methodology. We also thank the staff of the Scientific and Technological Services of the University of Barcelona for their help in the use of cell technologies.

### Funding
The authors received no funding for this work.

### Competing Interests
The authors declare there are no competing interests.

## Author Contributions

- Floriana Rotondo performed the experiments, analyzed the data, prepared figures and/or tables, reviewed drafts of the paper.
- María del Mar Romero conceived and designed the experiments, performed the experiments, reviewed drafts of the paper.
- Ana Cecilia Ho-Palma performed the experiments, reviewed drafts of the paper.
- Xavier Remesar performed the experiments, analyzed the data, reviewed drafts of the paper.
- José Antonio Fernández-López analyzed the data, prepared figures and/or tables, reviewed drafts of the paper.
- Marià Alemany conceived and designed the experiments, analyzed the data, wrote the paper, prepared figures and/or tables, reviewed drafts of the paper.

## Animal Ethics

The following information was supplied relating to ethical approvals (i.e., approving body and any reference numbers):

The Committee on Animal Experimentation of the University of Barcelona authorized the procedures used in the present study.

The animals were only subjected to euthanasia under isoflurane anesthesia, this is not a procedure requiring special permission but simple verbal communication of the execution of the procedure to the Acting Director of the Animal House. The sacrifice of the animals was carried out within a periodic culling procedure to reduce the population of the animal room. The animals were simply used for tissue sampling after anaesthesia instead of leaving them to die because of overdose of anaesthesia. No other manipulation was done on the animals, in accordance with the rules established by the Committee.

## Data Availability

University of Barcelona Repository http://hdl.handle.net/2445/102243.

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
