# Peer review of "Quantitative analysis of rat adipose tissue cell recovery, and non-fat cell volume, in primary cell cultures"

_PeerJ, doi:10.7717/peerj.2725_

## Round 0.1 · original submission · Minor Revisions

· Academic Editor

Minor Revisions

Dear Marià,

We would potentially be willing to publish this work as both reviewers felt there was value in your study by providing a very carefully controlled baseline for preparation and characterization of epididymal WAT deposits as well as more general interest in your findings of the cellular composition of the WAT, which I myself found quite interesting. However, some revision will be required as at the same time there was concern that the limited range of parameters tested (1 time point, 1 fat store and 1 functional assay) also limit this value. In particular I think that the concern that the only assay for cell functional viability was basal lactate production is significant as to gauge the quality of preps requires data on response to stimulus and readers will need this to judge your approach with regards to functionality of the final cell product. Thus it will be necessary to engage an additional assay testing the response to stimulation or suppression of lipolysis to make this acceptable for publication.

Additionally, though I will not require it as the reviewers did not make it mandatory, I would encourage you to simply test a small subset of your parameters at another age of the animals. As 18 weeks is well past sexual maturity one might not expect much change at 36 weeks and thus if there is little change in the measured parameters for example of Table 1 in 36 week old rats then other researchers could reasonably apply your findings across a wider age range of animals. If you do not wish to undertake this additional experiment it will be necessary to clearly state for readers this limitation of the study and the possibility that the parameters you have measured will differ drastically at other ages.

Finally, there are many specific points raised by reviewer 2, all of which seem reasonable and can be dealt with without additional experimentation. I would expect all of these to be addressed. In particular, I would highlight both the issues of experimental detail and statistics. For example regarding the former the reviewer asks for detailed information on the nylon mesh hose. In reading myself through the materials and methods, while I acknowledge it is happily more detailed than one generally finds, nonetheless the variability in the field in adipocyte isolation makes it critical that every single parameter be given. Thus please go beyond the specific requests of the reviewer to include for example not only collagenase catalogue number, but also how stored and length of time from resuspension if maintained in liquid form for any amount of time etc. Finally, reviewer 2 commented on the need for more statistics given as well as statistical description. I fully agree with this and would go a bit further in expecting in addition to explanation of what statistical tests are used that there be greater consistency within tables in types of error measured. I do not see a reason for alternating between standard error and standard deviation and unless one that makes good scientific sense can be clearly stated I would recommend just showing standard deviations.

Reviewer 1 ·

Basic reporting

No Comments.

Experimental design

No comments.

Validity of the findings

The largest issue is the applicability of the findings to use of rats of other ages, use of other WAT depots and finally inherent variability between labs in the isolation of primary cells. Indeed, even lot numbers of collagenase used can be an issue in reproducibility of adipocyte isolation. However, the present report serves as a baseline for other groups.
There is also concern about the measurement of cell viability. Isolated adipocytes are usually used day of preparation; prolonged culture (2 days) is standardly done as minced tissue pieces (without collagenase digestion) or use of an artificial matrix such as Matrigel. There is also no stimulation of the cells to an agonist or hormone; only basal production of lactate is measured.

Comments for the author

The authors have carefully documented the makeup and recovery of cell types during collagenase digestion of epididymal white adipose tissue from 18 week old rats. The experiments appear to have been carefully conducted and exhaustively analyzed, the data are clearly presented and the manuscript is clearly written and easy to follow. The major issue with the findings is that they represent a single "snap shot" - analysis of one WAT from rats of one age performed in one lab. However, the findings will serve as a baseline for results from other groups. A more minor concern is that basal lactate production was the sole parameter used to measure cultured cell viability as opposed to stimulation and/or suppression of lipolysis, the most physiologically relevant in vitro assay, so responsiveness to extracellular stimuli that regulate adipocyte metabolism were not studied/reported. These two limitations could be raised in the Discussion for a more comprehensive overview of the results.

·

Basic reporting

In general, the manuscript conforms to all of PeerJ’s guidelines for Basic Reporting. However, there are a few minor issues to address, as follows:

a) Abstract:
i. To conform to the PeerJ standard structure, headings in the abstract should be in bold, and each heading should begin a new paragraph.
ii. In the first line of the abstract, the word ‘disperse’ would be better replaced by the word ‘diffuse’, as this is a more appropriate adjective.
iii. The following sentence in the abstract could be re-written to improve clarity: “Studies with adipocytes often require their isolation from WAT breaking up the matrix collagen fibres, but primary cultures of these cells could not be easily correlated to intact WAT, since often recovery and viability are unknown.” I suggest instead writing, “Studies with adipocytes often require their isolation from WAT by breaking up the matrix of collagen fibres; however, it is unclear to what extent adipocyte number in such primary cultures correlates with their number in intact WAT, since recovery and viability are often unknown.” This will also clarify the rationale for your research question.
b) Introduction:
i. Line 39: please change “WAT sites” to “WAT depots”, because “sites” can also be used as a verb, which adds confusion to the original phrasing.
ii. Line 40: “protean” is a synonynm for “adaptable”, so the phrase “protean adaptability” is a tautology. I feel that removing the word “protean” would make your point clearer (i.e. use “adaptability” only, or perhaps “versatility” only).
iii. Lines 75-78: For clarity I recommend updating this sentence, for example to “This is further confounded by the direct estimation of cell number via DNA analysis, which (at least in mammals) would not detect the numbers of erythrocytes, but would detect numbers of small hematopoietic cells (Luche et al. 2015), marcrophages and lymphocytes (Sell & Eckel 2010). The latter non-adipocyte populations would then be counted as ‘adipocytes’, despite having a volume about 10^5-fold smaller.”
c) Materials and Methods:
i. Line 125: update to “…thin-floating lipid layer”.
ii. Line 129: update to “…the sum of the volume of the lower phase…”
d) Lines 358-360: The authors state, “The actual quantitation of the mass of adipocyte cytosol and its correlation with metabolic activity is a study worth carrying out, to definitively erase the assumption that WAT is basically an inert dump for excess energy.” I have two issues with this: firstly, few researchers in the field of metabolism and endocrinology still view WAT as “an inert dump for excess energy”, as there is now a broad consensus of its many other functions (e.g. endocrine and beyond); secondly, I think it is improper to say that an experiment should be done to show something, as this suggests that you have already made up your mind about the outcome (i.e. here you are assuming that the cytosolic mass would positively correlate with metabolic activity, which might not be the case). Instead I suggest that you re-phrase this, perhaps to “The actual quantitation of the mass of adipocyte cytosol and its correlation with metabolic activity is a study worth carrying out, to further test to what extent the cytosolic components of adipocytes might influence broader metabolic activity.”

Experimental design

The research question and gap in knowledge is clearly defined, and the Methods are generally well described. There are a few minor issues that should be updated as follows:

a) Line 116: If possible, please provide further details for the nylon mesh hose, i.e. manufacturer, product number, and pore size (µm) of the mesh.
b) Line 118: Please give further details of the plastic syringes, i.e. were these polystyrene, polypropylene, or another type of plastic? This is relevant as adipocytes adhere with different degrees to different types of plastic. If possible, please provide the manufacturer and product number.
c) Line 123: Please indicate the volume of each aliquot of the cells’ fraction used for incubation.
d) Line 126: Please indicate the temperature of the room (approximately), as this can vary substantially between labs.
e) Line 206: “Each well received 400 µL of the adipocyte suspension…”; did the authors calculate cell concentration before adding the cells? If so, this should be stated here.

There are also two more important issues that the authors should comment on and perhaps address in the manuscript:

f) Lines 137-139: The authors state that, in WAT, only adipocytes contain sufficient lipid to float. However, macrophages in adipose tissue can also accumulate lipid (http://www.ncbi.nlm.nih.gov/pubmed/21266330) and have been reported to account for 10-17% of all cells in the floated fraction of collagenase-digested WAT (http://www.ncbi.nlm.nih.gov/pubmed/21545734). Please could the authors comment on this, i.e. might their measurements be confounded by including macrophage lipid content? This is also relevant to line 260 and line 265, where the authors state, “We assumed that practically all WAT fat was present only in adipocytes…” and “there were no other fat-carrying cells in the system.”
g) In Figure 1, how did the authors determine that these are adipocytes, and not simply free lipid droplets? For example, were cells stained with DAPI or Hoescht to confirm the presence of nuclei in each cell? This is important to address, given that much lipid is lost during the adipocyte isolation (i.e. Table 2).

Validity of the findings

Generally the Results and Discussion are clearly and logically presented. One major limitation is the lack of statistical analyses presented to back up the findings. The differences are often large (e.g. between the numbers of the different cell types), and therefore presumably would achieve statistical significance; however, the manuscript should still present a statistical analysis of the findings. Indeed, in the Materials and Methods it is stated that statistical analyses were carried out (line 229), but no such analyses are presented in the manuscript. This must be done. The nature of the statistical tests used must also be stated in the Methods section, or in the legends to figures or tables, when appropriate.

Another issue is with the key conclusion that the ‘live cell’ volume of adipocytes in WAT is only around 3% of total tissue volume, i.e. much smaller than expected and much lower than in other tissue types. One concern here is that, using the Folch extraction method, some membrane lipids may also be extracted, which would result in over-estimation of the amount of fat in the intracellular lipid droplet. If so, this would under-estimate the “live cell” volume of the adipocytes. Please could the authors comment on this possibility? In the Methods section (lines 148-149) they state, “…most of the membrane lipids were not extracted”; however, it is not clear if the authors confirmed this empirically.

Comments for the author

I enjoyed reading your manuscript, which is the most in-depth analysis of WAT cellularity that I have read. It is indeed very surprising that erythrocytes make up the majority of cells in WAT, and I find it encouraging that, when lipid volume is excluded, adipocytes still comprise a greater proportion of total WAT volume than do the stromal cells (perhaps emphasising the importance of adipocytes in WAT, despite their relatively lower number than stromal cell types). Thank you for your efforts in conducting these analyses; I hope that you will be able to address the points that I have raised, as I feel that doing so would further improve the manuscript.

---

## Round 0.2 · Minor Revisions

· Academic Editor

Minor Revisions

The two core points that the second reviewer has noted were both raised in the initial reviews and so I think it is very reasonable to expect that they be addressed to the reviewer's satisfaction. Therefore, I would encourage you to carefully and completely make the changes requested to your revision. I would note that when the initial decision is for minor revisions that before sending a manuscript back to the reviewers I check the rebuttal against the revised and previous manuscript to see if I can make a final decision without having them re-review the manuscript and in doing so I also felt that these issues had not been fully addressed. Moreover, as I am well aware that different experiments require different statistical tests, I also ran this aspect by a bioinformatician colleague who agreed with the second reviewer that it was not clearly stated in the text why a particular test was used and that with only 4 rats the more relevant statistic for most of what has been done here would be SD. As both reviewers will be in favor of publication pending these changes I would strongly encourage you to make them, especially as they only require some minor calculations from existing data and adding a few sentences here and there to make clear to the reader the potential limitations due to not staining for nuclei and explaining throughout why which statistical tests were used. If you can make these changes there will be no need for an additional detailed rebuttal letter and please submit in addition to the revised text file one with marked changes for just these latest changes so that we can quickly assess that the issues re-raised by the reviewer have been addressed.

Reviewer 1 ·

Basic reporting

Fine

Experimental design

Solid

Validity of the findings

Acceptable

Comments for the author

Previous concerns have been addressed by the authors.

·

Basic reporting

I thank the authors for addressing my original comments, either by incorporating the suggestions into their revised manuscript, or for making a logical rebuttal to the points that I raised. I’m grateful for the thoughtful response to point ‘d’; although I still prefer stating that experiments are done to ‘test’ ideas rather than to ‘show’ a particular result, the authors clearly have considered the reasons for their phrasing and therefore I am happy with this.

Experimental design

As above, I’m thankful to the authors for addressing the previous points raised in response to the original submission. Regarding the ‘minor points’ that I previously raised (‘a’ – ‘e’ in my previous response) please could, the authors make the following minor update:

a) Lines 223-224: “Each well received 400 μL of the adipocyte suspension (a second 100 μL aliquot was taken simultaneously for counting)”. Thank you for indicating that the cell concentration was counted; to make this absolutely clear, please could you update this sentence to ““Each well received 400 μL of the adipocyte suspension (a second 100 μL aliquot was taken simultaneously to determine adipocyte concentration)”.

The authors’ response to my previous point ‘f’ is very comprehensive, presenting a logical argument that there will be minimal non-adipocyte contamination in the floating layer. The revised manuscript is much improved by including these details in lines 146-154. However, having argued this, it is very odd that the authors then respond to my point ‘g’ (use of nuclear stains to distinguish adipocytes from free lipid) by stating “We… did not consider the measurement of nuclei, in part because of the more than probable possibility that some macrophages were attached to the adipocytes (thus making the specific adipocyte measurements inviable).” So, on the one hand the authors are saying that the presence of contaminating cells is unlikely, but they then argue against nuclear stains by stating that it is ‘probable’ that some macrophages are attached to the adipocytes.

It is also argued that the orientation of the cells might hide the nucleus, preventing it from being detected. However, in my experience the fluorescence of a DAPI-stained nuclei is still visible, in floating adipocyte preparations, even when the nucleus faces away from the objective (perhaps because some residual fluorescence passes through the cell).

In the rebuttal it is also stated that “The use of UV-fluorescent dyes seem the obvious solution, but then the macrophage nuclei could not be distinguished from those of the adipocytes and the shape and size of the latter could not be established, preventing its counting…”. But, in the response to my previous point ‘f’, the authors state “the size of a macrophage full of fat is at most three orders of magnitude smaller than that of a mature adipocyte.” Therefore, if there were any contaminating macropaghes in the adipocyte preparations then these should be easily distinguishable based on their much smaller size.

I fully support their approach of allowing the lipid “droplets to coalesce in the fat layer. This may remove the larger ones, including those with the size of most adipocytes.” In my lab we also do this to remove free lipid; it generally works well. However, I do not agree with the authors’ arguments against using fluorescent nuclear dyes to distinguish adipocytes from lipid droplets, as in my experience this provides a more objective method of excluding free lipid from the final analysis. If there were any remaining concern about including non-adipocyte populations (e.g. macrophages), then other stains could be used for this purpose (e.g. staining macropahges with CD11b, CD45, etc).

In any case, the authors make good arguments that most of the cells in the floating layer will be adipocytes, and that most free lipid will be removed by allowing it to coalesce before counting the adipocytes. Therefore I do not insist on further experiments that incorporate the nuclear staining approach. However, readers should be made aware that the method might be improved by staining for nuclei (and perhaps other cell markers) in the preparations of floating cells. Therefore, please could the authors add one or two sentences to the Discussion to highlight this?

Validity of the findings

a) I’m grateful for the authors’ response to my concerns about statistical analysis. I agree that there is no need for statistical comparison of some parameters (e.g. comparing sizes of adipocytes with those of erythrocytes, which we know will differ greatly). I also agree strongly that the obsession with ‘statistical significance’ of P < 0.05 is rather arbitrary and in many ways is damaging science. However, in the rebuttal it is stated, “There are no comparisons because this is not a study in which we compare entities…” and “We studied a single WAT depot, in a single animal model (using four animals, in any case), what could we compare with?” I disagree: a major conclusion of the study is that erythrocytes make up the majority of total cell number in WAT. This seems likely based on the finding (Table 1) that erythrocytes represent 73.7 +/- 11.8 % of all stromal cells; however, the obvious statistical comparison here is between the numbers of adipocytes, nucleated stromal cells and erythrocytes. These could each expressed as the % of total cells in WAT and, across the four rats, the mean and SD of each cell % could be determined. The mean % of each cell type could then be compared by one-way ANOVA to determine if, as suspected, the % of erythrocytes is significantly greater than that of the other cell types (adipocytes and nucleated stromal cells). By normalising to total cell number you will negate the variability between the rats, which in your rebuttal you highlight as one concern.

Related to this, the Editor raises the issue of the use of SD vs SEM. I agree with the Editor that the biological variability (between animals) should be indicated, as SD, in the Tables and Figures. In the rebuttal it is stated, “However, when in a single measure (such as the counts of cells, cell sizes, etc., a single sample contains hundreds of replicas, which made the sem values excessively small to show the actual variability (exactly the opposite for the case exposed above). If the data from four animals are counted together, the sem data go even lower, even arriving to show significant differences (i.e. using a simple test such as Student's t) even between each individual animal with respect to the others.” Here I think it would be inappropriate to combine the numbers of counts within each animal (i.e. for cell size) with the numbers between each animal, as you are then artificially multiplying the number of independent biological replicates (which in this case would correspond to the four animals, rather than the many 100’s or 1000’s of cells counted within each animal). Instead I recommend taking the mean (or median, if not normally distributed) counts for each animal (e.g. mean/median cell size), and then calculating the mean and SD of these across the four animals.

It is unclear to me why the data in Table 1 include SD, but much of the data in Tables 2-5 do not. I strongly recommend that the authors calculate each of these parameters for each animal, and then in the tables present the mean +/- SD of these for the four animals. Knowing the inter-animal variance will be essential to readers because, if they wish to use this method in their own experiments, knowing the expected variability will be crucial for doing power calculations to decide sample sizes. I feel that this update should be relatively straightforward as it requires only further analysis of the existing data, rather than any additional experiments.


b) Thank you for addressing my concerns regarding the potential contribution of membrane-derived lipids. Your reasoning is very clear, and the revised manuscript more clearly reflects this (i.e. updates made in lines 280-287). The calculations given in your rebuttal provide an even stronger case against a major contribution of membrane lipids, so I wonder if it is worth including these calculations in the manuscript? Or, for brevity, perhaps update line 287 with “…there were no other fat-carrying cells in the system in mass and/or numbers sufficient to alter the results. Our previous work provides additional calculations that further support this conclusion (Remesar et al., 2015).”


c) One new (minor) point concerns the new data on glycerol and NEFA production. On lines 306-308 it is stated, “The higher rates of glycerol efflux in comparison with those of NEFA prove by themselves that most of glycerol was not of lipolytic origin…”. However, it is notable that, while adipocytes recycle NEFA (i.e. re-uptake), re-uptake of glycerol is only minimal because glycerol is instead used in the liver as a substrate for gluconeogenesis (http://www.pnas.org/content/102/31/10759.full). Therefore, their observation of greater glycerol vs NEFA concentrations does not prove that most of the glycerol is not of lipolytic origin. The authors should perhaps revise their statement to reflect this.

Comments for the author

Thank you again for the opportunity to review your manuscript, and for your thoughtful replies to my initial comments. Your revisions have clearly improved the original submission, and I hope that the few remaining issues can be addressed.

---

## Round 0.3 · accepted · Accept

· Academic Editor

Accept

Thank you for addressing the second round of revisions. While I know that it is unusual to undergo multiple rounds of revision, I think it is important to make the paper as clear to readers as possible. In this case I think that the additional qualifications have made some of the points you made eloquently in the rebuttal now also clear to the readers and this has strengthened the paper accordingly. Best, Eric